# SiON_x_ Coating Regulates Mesenchymal Stem Cell Antioxidant Capacity via Nuclear Erythroid Factor 2 Activity under Toxic Oxidative Stress Conditions

**DOI:** 10.3390/antiox13020189

**Published:** 2024-02-01

**Authors:** Neelam Ahuja, Kamal Awad, Su Yang, He Dong, Antonios Mikos, Pranesh Aswath, Simon Young, Marco Brotto, Venu Varanasi

**Affiliations:** 1Bone-Muscle Research Center, College of Nursing and Health Innovation, University of Texas at Arlington, Arlington, TX 76010, USA; 2Department of Material Science and Engineering, University of Texas at Arlington, Arlington, TX 76010, USA; 3Department of Chemistry and Biochemistry, University of Texas at Arlington, Arlington, TX 76010, USA; 4Center for Engineering Complex Tissues, Center for Excellence in Tissue Engineering, Rice University, Houston, TX 77005, USA; 5Department of Oral and Maxillofacial Surgery, University of Texas Health Science Center at Houston, Houston, TX 77054, USA

**Keywords:** silicon oxynitride, reactive oxygen species, plasma-enhanced chemical vapor deposition, antioxidant activity, osteogenic differentiation

## Abstract

Healing in compromised and complicated bone defects is often prolonged and delayed due to the lack of bioactivity of the fixation device, secondary infections, and associated oxidative stress. Here, we propose amorphous silicon oxynitride (SiON_x_) as a coating for the fixation devices to improve both bioactivity and bacteriostatic activity and reduce oxidative stress. We aimed to study the effect of increasing the N/O ratio in the SiON_x_ to fine-tune the cellular activity and the antioxidant effect via the NRF2 pathway under oxidative stress conditions. The in vitro studies involved using human mesenchymal stem cells (MSCs) to examine the effect of SiON_x_ coatings on osteogenesis with and without toxic oxidative stress. Additionally, bacterial growth on SiON_x_ surfaces was studied using methicillin-resistant Staphylococcus aureus (MRSA) colonies. NRF2 siRNA transfection was performed on the hMSCs (NRF2-KD) to study the antioxidant response to silicon ions. The SiON_x_ implant surfaces showed a >4-fold decrease in bacterial growth vs. bare titanium as a control. Increasing the N/O ratio in the SiON_x_ implants increased the alkaline phosphatase activity >1.5 times, and the other osteogenic markers (osteocalcin, RUNX2, and Osterix) were increased >2-fold under normal conditions. Increasing the N/O ratio in SiON_x_ enhanced the protective effects and improved cell viability against toxic oxidative stress conditions. There was a significant increase in osteocalcin activity compared to the uncoated group, along with increased antioxidant activity under oxidative stress conditions. In NRF2-KD cells, there was a stunted effect on the upregulation of antioxidant markers by silicon ions, indicating a role for NRF2. In conclusion, the SiON_x_ coatings studied here displayed bacteriostatic properties. These materials promoted osteogenic markers under toxic oxidative stress conditions while also enhancing antioxidant NRF2 activity. These results indicate the potential of SiON_x_ coatings to induce in vivo bone regeneration in a challenging oxidative stress environment.

## 1. Introduction

High-energy traumatic bone injuries, infections requiring surgical debridement, and large tumor resections result in large bone defects that are often associated with compromised wound healing resulting from deficient vascularization, hypoxia, wound contamination, or chemoradiotherapy [1,2]. These factors further complicate the defect healing process due to the accumulation of reactive oxygen species (ROS) and prolonged inflammation which may not be eliminated by the protective antioxidant mechanisms in the body [3,4,5]. This accumulation of excessive toxic radicals of ROS leads to oxidative stress which increases osteoclastic activity and disrupts the bone remodeling process [4,6]. Prolonged oxidative stress causes damage to the nuclear acids and proteins, causing irreparable cellular injury and restricting cell viability, growth, and proliferation. Exogenous or dietary antioxidants have been shown to reduce ROS levels and increase osteocalcin activity via increased antioxidant activity [3,7,8,9] (e.g., nuclear erythroid factor 2 (NRF2), superoxide dismutase (SOD1), and glutathione peroxidase (GPX)). Yet, these exogenous approaches are unable to overcome the challenges of spontaneous healing and completely regenerating large bone defects [10]. Further, an alternative approach of using the local delivery of small-molecule antioxidant agents in biomaterials can intrinsically induce rapid cell recruitment, maintain viability, and provide the surface needed to support healing. Thus, antioxidants play a central role in lowering ROS levels while inducing angiogenesis and osteogenesis.

It has been established in pertinent literature that NRF2 is a key transcriptional factor responsible for activating an antioxidant response reaction against oxidative stresses [11,12]. NRF2 can be promoted during a traumatic injury or fracture, which induces a defense mechanism towards oxidative stress and regulates the bone healing rate [13,14]. NRF2 plays a vital role in promoting bone and vascular healing by enhancing cell viability and inducing migration, endothelial cell angiogenesis, and mesenchymal stem cell osteogenesis [15,16]. NRF2 also acts as a master antioxidant promoter that reduces ROS levels to promote healing. NRF2 is activated by two key mechanisms: (1) the phosphorylation of cell surface glycogen synthase kinase (p-GSK3-beta), which activates the promoter regions of the NRF2 gene, and (2) the presence of electrophilic compounds, such as sulforaphane, which can indirectly affect Kelch-like ECH-associated protein (Keap1)–NRF2 nuclear binding through the modification of cysteine residues on the Keap1 surface which subsequently affect NRF2 activity [11,14,16,17].

Current treatment options for large and compromised bone defects involve the use of fixative implants such as titanium implants for functional reconstruction. However, these materials are not bioactive and lack antioxidant properties [18,19], and their use sometimes leads to microbial biofilm formation, causing infection [20]. The consequent poor osteointegration, implant loosening, and failure further compromise the healing process [21,22]. Various coating materials are being actively studied to overcome these disadvantages [23,24,25,26,27]. Yet, failure due to thermal expansion mismatch [28,29,30,31], poor coating quality owing to high temperatures (>700 °C), reduced bioactivity [32,33], and reduction in osteogenic activity [34], resulting in immature bone healing and fibrous tissue attachment [35] and poor long-term stability [36], are problems which still need to be addressed. New biomaterials or coatings on already existing biomaterials/fixative materials that can precisely enhance the wound healing environment are thus urgently needed for volumetric bone repair.

Our previous studies have shown that SiON_x_ and silicon oxynitrophosphide (SiONP_x_) coatings induce antioxidant activity that reduces ROS levels and inflammation, resulting in early bone tissue regeneration [37,38,39,40,41]. Similarly, silica (SiO_2_)-based nanoparticles/surface modifications have been widely studied for their antibacterial properties [42]. SiON_x_ films formed by plasma-enhanced chemical vapor deposition (PECVD) have been applied to a myriad of substrates, including Si wafers [37,40], metal devices such as titanium devices [43], and even ceramics and polymers [44]. We have conducted such studies and have found that irrespective of the substrate type, the coatings always maintain an amorphous structure, and their binding chemistry forms a covalent/ionic bond with the substrate, as measured by X-ray diffraction, X-ray absorption near-edge structure (XANES) analysis, and electron microscopy analysis in microstructure and morphology studies [37,38,39,40,41,43,45,46]. SiON_x_ is an amorphous material with varying levels of tetrahedral and trigonal chemical bond structure depending on the N/O ratio within the films [41]. This ratio is controlled by the material formation from source gases NH_3_ and N_2_O under a reductive ionized gas environment [41]. This approach is advantageous as the coating thickness, atomic ratio, and interfacial formation on a biomedical device can be precisely controlled [32,43]. Further, these materials can be fabricated at relatively low temperatures (<400 °C), due to which the thermal expansion mismatch between the implant substrate and the coating layers is markedly reduced. In addition, the film can be formed relatively quickly (within 1 h), simplifying the “made-to-order” or “patient defect tailored” manufacturing process. These bespoke SiON_x_ nanolayered coatings on biomedical devices will lead to the targeted delivery of the required antibacterial and antioxidant effects to the defect/injury site.

Their application in our previous studies led to a reduction in the lipid peroxidation on the SiON_x_-coated implants along with an increase in the angiogenic activity [37]. The silicon oxynitride chemical structure contains a range of tetrahedral bonds with a low N/O ratio and trigonal bonds with a high N/O ratio. We previously demonstrated that the random bonding model (RBM) is valid for a low N/O ratio, whereas the random mixture model (RMM) is applicable for a high N/O ratio. We also showed that increasing the N/O ratio lowers the energy required for 2p to 3d electron transitions. The presence of 3d electron orbits is needed to stimulate antioxidant activity, which is spin-orbit-restricted and only becomes activated by transition metals with readily available 3d electrons (Zn, Cu, Mn, Fe). More fundamentally, an increase in the N/O ratio changes the partial charge and the surface dipole that affects how the surface will share electrons with attached cellular membranes in which biomolecules depend on the energy state of these surface electrons.

Hence, we aim to study the SiON_x_ surface and the Si ion release effects on bacterial growth as well as MSC proliferation and differentiation in vitro with and without oxidative stress conditions to understand the underlying molecular mechanism. To achieve this objective, we use increasing nitrogen-to-oxygen ratio (N/O) in the Si−O−N composition as the independent variable. In this context, the N/O ratio essentially describes the solid-state substitution of O for N as it is added into the material and is dictated by the NH_3_/N_2_O gas flow in PECVD. The N/O ratio affects the local surface dipole that plays a role in the oxidative state (or valence state) of the surface. Subsequently, cells exposed to these surfaces with differing oxidative states can exhibit differences in viability, proliferation, and differentiation due to the ability of the surface to potentially affect how the cell interacts with its local environment. This effect can also be observed with biomaterials tested with osteogenic progenitors that can induce marker activity, such as alkaline phosphatase, as early as 1–7 days after exposure [47]. Thus, a detailed examination of these markers is needed to ascertain the full effect of the impact of these biomaterials.

Here, we study the effect of increasing nitrogen (N) content in SiON_x_ surface coatings on the bacteriostatic effect, antioxidant activity, ROS, and osteogenesis. Finding the effective N/O ratio in a SiON_x_ coating will enable the development of engineered coatings for fixative implantable materials with specific surface chemistry capable of triggering beneficial biological activity for faster bone healing. In the present study, attention is given to antioxidant activity maximization and thus enhancement in oxidative stress reduction. We propose that SiON_x_ induces antioxidant activity and reduces oxidative stresses through the NRF2 pathway, and thus, we aim to uncover the role of the NRF2 signaling pathway in the SiON_x_ antioxidant activity.

## 2. Materials and Methods

### 2.1. Fabrication of the Coatings

SiON_x_ and SiONP_x_ thin film coatings were fabricated using the PECVD technique according to our previously published methods [37,39,40,41,43,45,48]. Briefly, the surface was prepared for the coatings by applying standard cleaning procedures based on the piranha solution (3:1 mixture of sulfuric acid (H_2_SO_4_, 96%) and hydrogen peroxide (H_2_O_2_, 30%)) for 10 min and was subsequently rinsed in deionized (DI) water for 1 min. Then, a TRION ORION II PECVD/LPECVD system (Trion Technology, Clearwater, FL, USA) was used to deposit a 1000 nm thin film coating of SiON_x_ or SiONP_x_ on a silicon wafer. Increasing the N_2_O gas flow rate increases the oxygen incorporation in the coatings relative to the nitrogen content. Thus, five different SiON_x_ chemistries were fabricated based on changing the N_2_O flow rate. The refractive index (*n*) of the deposited coatings was measured and was used to distinguish the coatings in this study. Refractive index values ranged from *n* = 2 for N_2_O = 0 standard centimeters cubed per minute (sccm) to *n* = 1.45 for N_2_O = 160 sccm. The study groups were as follows: (1) positive control (tissue culture plate surface), (2) *n* = 1.45, (3) *n* = 1.57, (4) *n* = 1.65, (5) *n* = 1.82, and (6) *n* = 2. The same methods were used for obtaining SiONP_x_, with the exception of phosphine gas (PH_3_) as a phosphorus source, as shown in Table 1. The composition of the coatings, the exact flow rate, and the refractive index of each coating are provided in Table 1.

### 2.2. In Vitro Studies

Human bone marrow mesenchymal stem cells (MSCs) were obtained from Lonza (Lonza Walkersville Inc., Walkersville, MD, USA). All cells were authenticated, performance assayed, and tested negative for mycoplasma, according to the provider. MSCs (passages 2–4) were grown according to the Lonza protocol in MSCGM BulletKit^TM^ (Lonza, PT-3238 and PT-4105) specific growth medium at 37 °C and 5% CO_2_ in a completely humidified incubator. For studying the effect of increasing nitrogen concentration in the coatings, the following groups (*n* = 4) were used for all in vitro studies: I—tissue culture plate (TCP); II—SiON_x_ *n* = 1.45; III—SiON_x_ *n* = 1.57; IV—SiON_x_ *n* = 1.65; V—SiON_x_ *n* = 1.82; and VI—SiON_x_ *n* = 2. Cells (5000–6000 cells/cm^2^) were seeded on TCP as control and SiON_x_ with different refractive indices and were allowed to grow for 7 days. All samples were cleaned using 100% ethanol followed by dry heat sterilization prior to each experiment [49].

In vitro cytotoxicity studies were performed according to our previously reported protocols [43,45]. Cytotoxicity and cell adhesion, viability, and proliferation were studied at 1, 4, and 7 days. The MTS-CellTiter^96®^ Aqueous One Solution Cell Proliferation Assay (Promega, Madison, WI, USA) was used for the quantitative analysis of cell growth and proliferation. The LIVE/DEAD™ Viability/Cytotoxicity Stain Kit (Thermo Fischer Scientific Inc., Waltham, MA, USA) was utilized for the qualitative analysis of cell adhesion and viability on the tested surfaces. Both assays were performed according to the manufacturer’s instructions and our previously published protocols [37,45]. The colorimetric absorbance of the MTS assay was determined using a microplate reader (SpectraMax^®^ i3, Molecular Devices, San Jose, CA, USA) at 490 nm. Normal and oxidative cell culture media without cells were used to account for any background that is subtracted from all MTS absorbance results. Live/dead fluorescence images were then taken using a DMi8 inverted Leica microscope (Leica Microsystems Inc., Deerfield, IL, USA), with green fluorescent calcein-AM for live cells and red fluorescent ethidium homodimer-1 for dead cells.

To induce osteogenic differentiation of the MSCs, hMSC differentiation BulletKit™—osteogenic was acquired from Lonza, PT-3002; it contains osteogenic differentiation basal medium and hMSC osteogenic SingleQuots™. All groups described above were used for all differentiation studies, including TCP as the control and SiON_x_ with different refractive indices. Differentiation studies were conducted for 7 days, with media collected for analysis on days 1, 4, and 7.

To study the effect of SiON_x_ on the osteogenic differentiation of MSCs, an Alkaline Phosphatase (ALP) Kit (Colorimetric) (Abcam, Waltham, MA, USA) was used to quantify the ALP activity in the collected media for differentiated MSCs after 1, 4 and 7 days.

Human umbilical vein endothelial cells (HUVECs) (passages 2 to 4), endothelial cell growth medium 2 (EGM-2) (used for cell growth and subculture), and endothelial cell basal media 2 (EBM-2) (used for the proliferation and differentiation experiments) were acquired from Lonza^®^. These cells were seeded at a concentration similar to that of the MSCs and were allowed to culture for several days prior to lysis.

After 7 days of differentiation, RNA was collected from the differentiated MSCs, purified using the miRNAeasy MINI KIT from QIAGEN^®^, and converted to cDNA using the Goscript^TM^ Reverse transcriptase kit from Promega Corporation. Reverse transcription polymerase chain reaction (RT-PCR) was performed using the TaqMan^®^ Gene Expression Assay with the standard protocol adjusted for 40 cycles. The obtained results were expressed relative to the housekeeping gene GAPDH and compared to the control (TCP), and the delta Ct method was used for calculations. Osteogenic markers RUNX2 (Runt-related transcription factor-2), BGLAP (osteocalcin, also OCN), ALP (alkaline phosphatase), and SP7 (Osterix) were assessed.

For OCN detection by immunohistochemistry, after 7 days of differentiation, the cell culture well plates were washed with PBS, fixed, and blocked with 10% goat serum. After blocking, the primary antibodies (Rabbit polyclonal OCN 1:500) were placed on the slides and incubated overnight at 40 °C. Then, Alexa Dye 594 Goat anti-Rabbit secondary antibodies in 1:200 dilution were introduced, and the sample was incubated for 1–2 h, washed twice with PBS, stained with DAPI for 15–20 min at room temperature, and evaluated under a DMi8 inverted Leica microscope (Leica Microsystems Inc., IL, USA).

Hydrogen peroxide (H_2_O_2_) was used to induce oxidative stress in the cells. H_2_O_2_ has been used in various in vitro studies as a major product of oxidative stress, produced by swift conversion from superoxide. A preliminary study was conducted using different H_2_O_2_ concentrations (0–1 mM) to determine the optimal H_2_O_2_ amount for mimicking an oxidative stress environment deleterious for the cells. A 30% *w*/*w* of H_2_O_2_ with stabilizer stock solution was acquired from Sigma-Aldrich. Based on the obtained findings, we determined 0.2 mM as the H_2_O_2_ concentration that is deleterious for the MSCs.

All cell culture studies stated above—including the cell viability and proliferation (MTS and live/dead assay) along with differentiation studies (ALP assay, PCR, IHC)—were repeated under oxidative stress conditions by introducing 0.2 mM of H_2_O_2_ to the cell culture environment. Additionally, antioxidant markers—nuclear factor erythroid 2-related factor 2 (NRF2) and Keap1—were utilized for the oxidative stress study.

We acquired human NRF2 siRNA from Santa Cruz Biotechnology, Inc. (SC-37030), Dallas, TX, USA, to knock down NRF2 (NRF2-KD) in the MSCs. We followed the manufacturer’s protocol to perform NRF2 siRNA transfection. The MSCs were seeded in a 6-well plate as described above and were incubated until 60–80% confluence was achieved. We prepared two transfection solutions (A and B). For obtaining Solution A, 6 μL of siRNA duplex was added to 100 μL of siRNA Transfection Medium (sc-36868) for each transfection, and for Solution B, 6 μL of siRNA Transfection Reagent (sc-29528) was added to 100 μL of siRNA Transfection Medium (sc-36868) for each transfection. The two solutions were mixed gently, and the resulting mixture was incubated for 15–30 min at room temperature before adding 0.8 mL of siRNA Transfection Medium. Next, the seeded cells were washed twice and then incubated at 37 °C with the siRNA Transfection Reagent for 6–7 h. After transfection, the reagent was removed and replaced with normal growth medium. The cells were allowed to grow for 4 days prior to the PCR analysis. We also performed an internal control for the siRNA transfection which included a scrambled sequence that would not lead to the specific degradation of any known cellular mRNA.

As staphylococci account for up to two-thirds of all pathogens isolated from orthopedic implant infections, methicillin-resistant Staphylococcus aureus (MRSA) bacterial assays were conducted to study the bacterial growth on the implant surfaces. When using MRSA, the ASTM and ISO standards for In Vitro Test Methods for Antimicrobial Coatings on Medical Devices were followed [50]. These bacteria form a biofilm on the surface of a foreign device such as a metallic implant, making infection eradication without the surgical removal of the device impossible. A biofilm is a community of sessile bacteria encased in an extracellular matrix comprising water, microbial cells, nutrients, polysaccharides, DNA, and proteins. The biofilm provides a protective matrix around the encased bacteria and is highly resistant to host immune defenses and antimicrobials [51]. Thus, MRSA was used to evaluate the bacteriostatic effect of the studied coatings as potential surface coatings for biomedical metallic implants. For each experiment, a single colony from the inoculation plate was used with Mueller Hinton Broth (MHB) as the medium. A total of 10^5^ colony-forming units/mL (CFU/mL) was seeded onto the material surface with an inoculation loop, and the samples were dried to remove all liquid and then incubated at 37 °C for 12, 24, or 48 h. A live/dead bacteria assay kit (LIVE/DEAD BacLight Bacterial Viability Kit, Thermo Fischer Scientific Inc., Waltham, MA, USA) was used to stain the bacteria on the surfaces for 15 min at room temperature, after which the samples were washed with PBS and imaged under an EVOS M5000 Imaging System. ImageJ software (NIH, version 1.53k) was used to further analyze the results.

### 2.3. Statistical Methods

The results are presented in box plots showing the mean, standard deviation, and significance levels. One-way ANOVA (Tukey’s pairwise comparison) was conducted to compare the means of more than two groups at a significance level of *p* < 0.05, and image analysis was performed in ImageJ. OriginPro 2017, Past3, and Microsoft Excel 2016 Software were used for graphics and calculations.

## 3. Results

As mentioned above, we have several publications on the PECVD SiON_x_- and SiONP_x_-based coatings examined here, and we have established their amorphous nature with unique surface characteristics. Therefore, the chemical composition, structure, and surface morphology of the deposited coatings were characterized to ensure full compliance with our published data (see reference [41] for a comprehensive characterization of these coatings). The coatings were characterized using XRD, XANES, FT-IR, XPS, NRA-RBS, HR-SEM with EDX, and HR-TEM [37,38,39,41,43,48]. Thus, the work presented here focuses on the bacteriostatic effect and the molecular mechanism behind the antioxidant activity of these coatings.

### 3.1. Bacteriostatic Effect

We studied the bacterial growth and MRSA adherence on the various SiON_x_ chemistries compared to the titanium surface (positive control). The quantitative analysis of the live MRSA bacteria on SiON_x_-coated surfaces compared to titanium after 12, 24, and 48 h (calculated by ImageJ software) is shown in Figure 1. A significant decrease in cell viability and proliferation is evident in all SiON_x_-coated samples, as indicated by a decrease in green staining (live bacteria) and an increase in red staining (dead bacteria) at 48 h (Figure 2). (Appendix A represent bacterial live/dead stains at 12 and 24 h, respectively.) A significant decrease in bacterial counts at 12 h was noted on the high-N/O-ratio SiON_x_ (*n* = 2.0) surface, which exhibited the lowest bacterial counts amongst all SiON_x_ groups after 24 and 48 h. This finding is indicative of a bacteriostatic effect on the SiON_x_-coated surfaces inhibiting bacterial growth and proliferation compared to titanium.

### 3.2. Cell Viability and Proliferation under Normal Conditions

We studied the MSC viability, growth, and proliferation on various SiON_x_ chemistries to evaluate the effect of increasing the nitrogen content relative to oxygen while keeping the silicon content in the coatings fixed. We performed an MTT assay for quantitative analysis along with qualitative live/dead fluorescence analysis (Calcein AM and Ethidium homodimer) after 1, 4, and 7 days of seeding on the various SiON_x_ surfaces. As can be seen from Figure 3, there are no significant differences among the groups or between groups and the positive control (TCP surface) on days 1 and 4 of cell proliferation under normal conditions, suggesting normal growth. However, after 7 days, cellular proliferation decreases as the N/O ratio increases, which can be attributed to MSC detachment on the un-patterned base silicon wafer surface.

### 3.3. Osteogenic Differentiation with Increasing N/O Ratio in SiON_x_ Surfaces

We studied the alkaline phosphatase (ALP) activity on various SiON_x_ chemistries using the Alkaline Phosphatase Assay Kit (Figure 4) and the cell culture supernatant of the seeded MSCs undergoing osteogenic differentiation at 1, 4, and 7 days. No significant differences in the ALP activity in SiON_x_ *n* = 1.82 and *n* = 2 were noted when compared to the control (TCP) on days 1 and 4. On day 7, there was significantly less ALP activity in the media supernatant of the SiON_x_ samples as compared to the control, which was attributed to the decrease in the cellular numbers as noted above. However, there was a significant increase in the ALP activity as the SiON_x_ nitrogen content increased relative to oxygen on day 7 (the SiON_x_ *n* = 2 group exhibited the highest activity level).

Gene expression analysis (rt-PCR) (Figure 5) reveals an upregulation of osteogenic markers (osteocalcin, ALPL, and Osterix) with an increase in the N/O ratio in the SiON_x_ as compared to the control. There is >2–3× upregulation of osteogenic markers in SiON_x_ *n* = 1.82 and *n* = 2 as compared to the control and the other SiON_x_ chemistries. Figure 6 depicts a similar upregulation of the antioxidant markers (SOD1, NRF2, and GPX1) with the increase in the N/O ratio in the SiON_x_ as compared to the control. Although there is a 2-fold upregulation in the antioxidant activity, the values are not significantly different from the baseline (with the exception of GPX1 for SiON_x_ *n* = 2).

### 3.4. Determining Minimal H_2_O_2_ Concentration for Inducing Oxidative Stress in MSCs

Hydrogen peroxide has been used in extant studies to induce oxidative stresses in the cellular environment. To determine the minimal H_2_O_2_ concentration that would have a deleterious effect on the MSC growth, we tested the cell viability (MTT and live/dead assay) after adding H_2_O_2_ in various concentrations (0.2 mM, 0.4 mM, 0.6 mM, 0.8 mM, and 1 mM) to the culture media (0 mM served as the control). As shown in Figure 7, a significantly lower number of live cells (15–20% less) was present at 0.2 mM and 0.4 mM, while more than 60% reduction in cell survival was noted at 0.6 mM, and almost all cells died at higher concentrations. We removed the oxidative stress medium after 24 h and placed the surviving cells on the normal growth medium for 3 days. We observed that even after the removal of the initial oxidative stress, there was a sustained insult on the cell survival. Hence, we chose 0.2 mM as our final concentration as there was a significant effect of the toxic oxidative stress environment and, on further growing the cells after the initial insult, the cell survival rate exceeded 60% (Appendix A).

### 3.5. Effect of Increasing the Nitrogen Concentration in SiON_x_ Chemistry on the Cell Proliferation and Differentiation under Oxidative Stress Conditions

We further analyzed the effect of cell viability and growth by MTS assay and live/dead staining on MSCs when seeded on the various SiON_x_ chemistries under toxic oxidative conditions. Figure 8 shows that increasing the nitrogen content relative to oxygen (when silicon is kept constant) has a protective effect on the MSCs against the toxic oxidative stress, showing a 2-fold increase in the number of cells on *n* = 2 as compared to *n* = 1.45 and *n* = 1.57 and showing a comparable number of cells to that recorded when no hydrogen peroxide was present in the environment. PCR analysis performed after differentiating the MSCs in an osteogenic medium for 7 days after the initial exposure to the oxidative stress environment reveals an increase in the RUNX2 upregulation (Figure 9) in SiON_x_ *n* = 1.82 when compared to the baseline with oxidative stress as well as the other studied SiON_x_ chemistries, while being comparable to the control with no oxidative stress. There is also a significant upregulation in ALPL and osteocalcin with SiON_x_ *n* = 1.82 compared to the baseline with/without oxidative stress as well as the other studied SiON_x_ chemistries (Figure 9).

Florescent microscopy also reveals significantly enhanced osteocalcin (OCN) activity (Figure 10) on the SiON_x_ surface. Antioxidant markers (SOD1, NRF2, and GPX1) are significantly overexpressed in SiON_x_ *n* = 1.82, which correlates to the overexpression of the oxidative markers in the same groups when compared to the baseline with oxidative stress and other SiON_x_ chemistries, as well as the control group with no oxidative stress involved (Figure 11). Keap1 marker expression was increased compared to the control with no oxidative stress; however, no significant differences are observed within the groups. There was also an increase in the expression of antioxidant NRF2 activity (Figure 12) on the SiON_x_ surface compared to the control.

We observed > 70% transfection with NRF2 siRNA after 6 h of incubation. Next, to study the effect of silicon on NRF2 and antioxidant activity, we grew the MSCs in normal growth media for 4 days post-transfection and added 0.5 mM of Si ions. RT-PCR analysis after NRF2 KD revealed that the NRF2 activity remained suppressed even after the addition of Si ions (Figure 13). Antioxidant expression indicates that, in the absence of transfection, there is a 2-fold increase in the SOD1 and GPX1 expression (Figure 13). A similar effect is observed in the control transfection group (scrambled sequence with no effect on mRNA), although when NRF2 is knocked down, there is no increase in the antioxidant expression compared to the baseline (Figure 13). There are no significant differences between the no-transfection group and the NRF2.

We conducted further testing to determine the effect of the coating on certain aspects of the signaling pathway related to NRF2. We tested SiON_x_ and further modified the structure with a small amount of phosphate to obtain SiONP_x_ coatings (P/N = 0.1) to assess their effects associated with signaling and angiogenesis using MSCs and HUVECs. NRF2 is activated when it is translocated to the nucleus by Kelch-like ECH-associated protein (Keap1), where it separates and allows NRF2 to activate the downstream antioxidant signals [52]. This NRF2 activation is further enhanced by the presence of 2^+^ or 4^+^ cations [17,53,54]. In our first evaluation of PECVD SiON_x_ coatings, we found that SiON_x_ with N/O = 2 enhanced osteogeneses via SOD1-dependent activity [40,55]. We further observed that SiON_x_ (with surface Si^4+^) spurred cytosolic NRF2 (Figure 14A) while Keap1 remained in the nucleus (Figure 14B).

In our previous reports [43,48], we noted that adding phosphate enhances HA formation and Si^4+^ release. Thus, we added a small amount of phosphate to make SiONP_x_ coatings (P/N = 0.1) [39]. When we increased the P/N ratio to 1.0 (Figure 14D), NRF2 and angiopoietin increased in vitro (SiONP2, Figure 14C,D), as did the MSC alkaline phosphatase (ALP) activity (Figure 14E). Accordingly, our working hypotheses are as follows: (1) increased charge transfer from the surface Si^4+^ valence state catalyzes ROS reduction (Equations (1)–(4)); (2) surface Si^4+^ may enhance NRF2 activity due to Keap1 sensitivity to 2^+^ and 4^+^ cations [53], thereby allowing NRF2 release; and (3) an increased P/N ratio in SiONP_x_ increases Si^4+^ surface ion release for the activation of these antioxidant signaling pathways. We will comprehensively explore the effect of P addition in SiON_x_ coatings as a part of our future studies.

**Si^4+^ effect:** Equation (2) produces 3× more H_2_O_2_ than Equation (1), which causes Equation (3) to produce more H_2_O + O_2_ (Equation (4)).
M^2+^/SOD1 + O_2_^−^ 2H^+^ = M^2+^/SOD1 + H_2_O_2_ (M = Zn, Cu)(1)
Si^4+^/SOD1 + 3O_2_^−^ 6H^+^ = Si^4+^ SOD1 + 3H_2_O_2_(2)
Fe^3+^/CAT + H_2_O_2_ = Fe^3+^/CAT + H_2_O + ½O_2_(3)
Fe^3+^/CAT + 3H_2_O_2_ = Fe^3+^/CAT + 3H_2_O + 1.5O_2_(4)

## 4. Discussion

In this study, we investigated the effect of increasing the nitrogen content in SiON_x_ surface coatings on bacterial growth and in vitro mesenchymal stem cell viability, proliferation, and osteogenic capacity. In addition, we examined the effect of the coatings on MSCs under hydrogen peroxide exposure inducing cellular-level oxidative stress.

Our results indicate that SiON_x_ coatings have a bacteriostatic effect, inhibiting bacterial growth and proliferation (Figure 1 and Appendix A) as compared to a regular titanium surface. MRSA is the most common bacterial infection related to bone and joint injuries and is the most prevalent cause of hospital-acquired infections. As these SiON_x_ PECVD-based coatings are amorphous in nature [55], they have an ability to partially dissolve in the physiological environment and release Si ions as well as facilitate the formation of silanol surface groups (Si−OH) similar to a bioactive glass surface [34,56,57]. It is known that bioactive glass stimulates various biological responses when immersed in physiological fluids, causing the exchange of network-modifier ions and Si−OH surface groups [56]. These surface changes activate mechanisms that efficiently inhibit bacterial growth and thus promote the adhesion and contamination of implants. We speculate that the SiON_x_ coatings have the same antibacterial effect. In the present study, increasing the nitrogen content in the SiON_x_ chemistry enhanced the coating’s bacterial inhibition compared to the Ti surface and SiON_x_ with a low nitrogen content. Thus, the presented data suggest a possible antibacterial/bacteriostatic effect of SiON_x_ coatings compared to the Ti implants.

Reactive oxygen species (ROS) have been shown to play a role in aging and several pathological conditions, as well as traumatic injuries [58,59]. The role of ROS in bone metabolism is dual, considering its effects under physiological or pathological conditions [60]. Under physiological conditions, ROS production by osteoclasts helps accelerate the destruction of calcified tissue, assisting in bone remodeling. Under pathological conditions, such as bone fractures, pronounced radical generation occurs [4,5]. The inhibition of the activities of antioxidant enzymes, such as superoxide dismutase and glutathione peroxidase, was found to increase the production of superoxide by osteoclasts, leading to a delayed bone remodeling process [5]. Therefore, oxidative stress is an important mediator of bone remodeling and healing, whereas the compensatory antioxidant mechanism helps restore the physiological process [8,61]. Accordingly, increased free radical production overwhelms the natural antioxidant defense mechanisms, subjecting tissues to hyperoxidant stress and delaying bone healing. In addition, antioxidant administration might help in the acceleration of healing of fractured bones to some extent but does not guarantee a full physiological recovery in cases of large and compromised bone defects. Thus, biomaterials that can treat the defect by reducing ROS and stimulating osteogenesis within the bone defect can greatly improve the efficacy of such treatments in repairing the lost bone.

Under normal physiological in vitro conditions, we observed no significant differences in the various SiON_x_-coated surfaces as compared to a cell-culture-treated surface as a positive control, indicating favorable cellular viability and proliferation on the coated surfaces (Figure 3). Increasing the N/O ratio had a positive impact on the MSC osteogenic differentiation, manifesting as an increase in the ALP activity (Figure 4) and various other markers (Figure 5 and Figure 6). When oxidative stresses were introduced to the physiological cell culture environment, we observed significant differences in the viability and proliferation of the mesenchymal stem cells with the increase in the N/O ratio in SiON_x_. Increasing the N/O ratio resulted in an increasing protective phenomenon (attributed to the change in charge) towards the mesenchymal stem cells with no significant differences in cell numbers when compared to cells grown in a normal physiological environment (Figure 8). Under oxidative stress conditions, we noted an upregulation of the osteogenic markers and antioxidant markers in the SiON_x_ *n* = 1.82 group compared to the other groups. This upregulation was similar to the control with no oxidative stress conditions or even significantly higher for osteogenic markers such as ALPL and osteocalcin. As previously reported by our group [33], the Si−H and N−H bonds increase with the increase in N content in the Si−O−N structure. Moreover, the increased nitrogen reduces the partial charge of constituent elements and changes the bonding structure from random bonding to random mixing. We observed that SiON_x_ with the refractive index of *n* = 1.82 enhances bioactivity and maximizes the antioxidant effect against the oxidative stresses, exhibiting [Si–Si]–[Si–O]–[Si–N] bonds as compared to SiON_x_ *n* = 2 (which is nitrogen-rich, SiN_x_).

Several studies have shown that NRF2 is a key transcription factor that maintains homeostasis in bone cells, especially against oxidative stresses [41,62]. The findings yielded by these studies implicate the antioxidant pathway activation of NRF2 against detrimental oxidative stress along with the enhancement of osteogenesis in MSCs as being involved in the bone healing of fractures. NRF2 is also known to be of crucial importance in reducing the impact of degenerative bone diseases [63]. Under physiological conditions, NRF2 is negatively regulated by its cytoplasmic antagonist Keap1. However, ROS can deactivate Keap1 to prevent NRF2 degradation [37,63]. This leads to NRF2 being translocated and binding to ARE, which encodes antioxidant enzymes and cytoprotective proteins [11]. Our previous investigations demonstrated the expression of several antioxidants, including SOD1 and GPX, on endothelial cell lines [37]. In this study, we show the overexpression of NRF2 under oxidative stress conditions for SiON_x_ (Figure 11 and Figure 12) along with other antioxidant markers. When we blocked NRF2 (NRF2-KD), we did not observe the upregulation of the antioxidant markers with the silicon ions as before (Figure 13). Moreover, we observed the nuclear localization of Keap1 in MSCs cultured on the SiON_x_ surface (Figure 14). These findings indicate that our initial assumption that SiON_x_ induces the antioxidant properties through the NRF2 pathway is correct. We are of the view that this overexpression of NRF2 activity and relatively unaffected Keap1 activity (normalized to the cell number) (Figure 13 and Appendix A) indicate the activation of the NRF2-ARE pathway to induce antioxidant enzymes and cytoprotective proteins along with enhancing osteogenic capacity in the MSCs.

Complicated bone defects can originate from a myriad of situations, as the “hidden” aspect of oxidative stress can manifest as a result of a patient’s biological conditions, such as diabetes or aging. Thus, biomaterials that can treat the defect by reducing ROS and stimulating osteogenesis for MSCs within the bone defect can greatly improve the efficacy of such treatments in repairing the lost bone.

In conclusion, the SiON_x_ coatings examined as a part of this work displayed bacteriostatic properties and supported mesenchymal stem cell viability and proliferation. Increasing the nitrogen-to-oxygen ratio while keeping the silicon content constant led to an increase in osteogenic capacity under normal physiological conditions. When the NRF2 overexpression on the SiON_x_ coatings with the highest N/O ratio was activated under toxic oxidative stress conditions, a protective antioxidant effect was observed, inducing ROS reduction and robust osteogenic marker expression and enhancing osteogenic capacity.

## Figures and Tables

**Figure 1 antioxidants-13-00189-f001:**
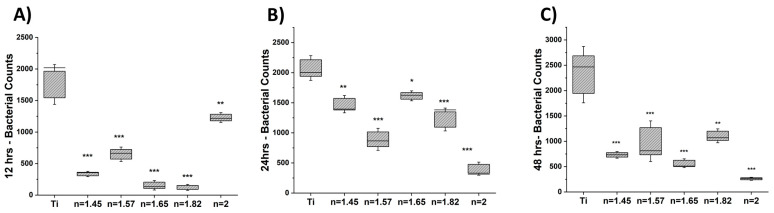
Bacteriostatic effect of SiON_x_ versus Ti. All SiON_x_ samples presented a significant decrease in the total number of bacteria at all time points: (**A**) 12 h, (**B**) 24 h, and (**C**) 48 h. At 12 h, the lowest bacterial counts were noted for SiON_x_ (*n* = 1.82), while the bacterial counts on SiON_x_ (*n* = 2.0) significantly decreased at 24 and 48 h. * *p* < 0.05, ** *p* < 0.01, and *** *p* < 0.001 present the significance compared to Ti as a control.

**Figure 2 antioxidants-13-00189-f002:**
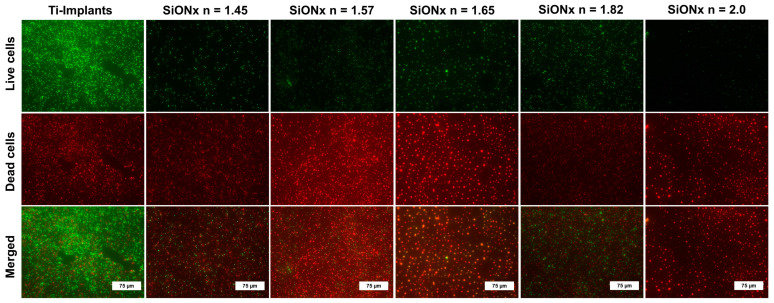
Bacteriostatic effect of SiON_x_ coatings versus Ti-based implants. Fluorescence images show the live (green) and dead (red) bacteria on different SiON_x_ surfaces compared to Ti-based implants after 48 h.

**Figure 3 antioxidants-13-00189-f003:**
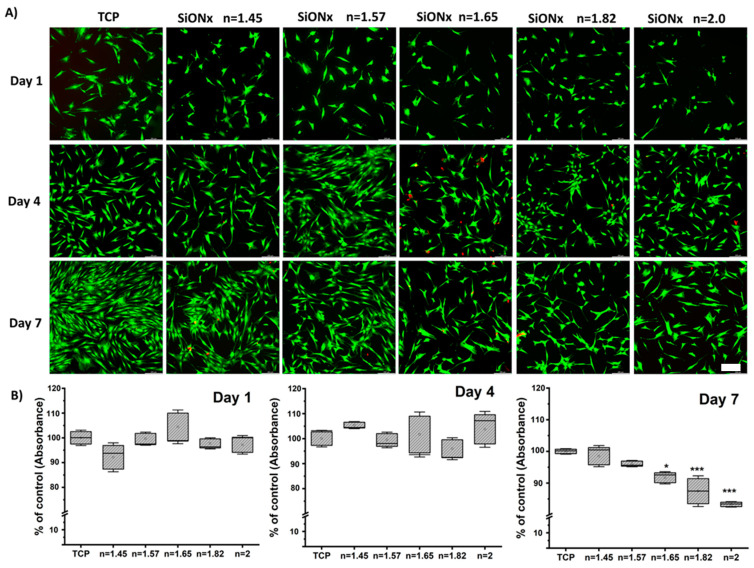
Cell viability and proliferation assay evaluated by (**A**) fluorescent live (Calcein AM) (green stain) and dead (Ethidium homodimer) (red stain) staining along with quantitative evaluation by (**B**) MTS assay after 1, 4, and 7 days of seeding on the various SiON_x_ surface chemistries. * *p* < 0.05 and *** *p* < 0.001 present the significance compared to the normal TCP as a control. Scale bar = 200µm.

**Figure 4 antioxidants-13-00189-f004:**
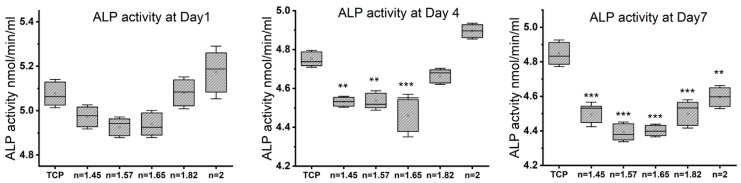
Alkaline phosphatase assay on days 1, 4, and 7 of osteogenic differentiation in MSCs on different SiON_x_ chemistries. The ALP activity in SiON_x_ *n* = 1.82 and *n* = 2 is not significantly different when compared to the control on days 1 and 4. ** *p* < 0.01 and *** *p* < 0.001 present the significance compared to the normal TCP as a control.

**Figure 5 antioxidants-13-00189-f005:**
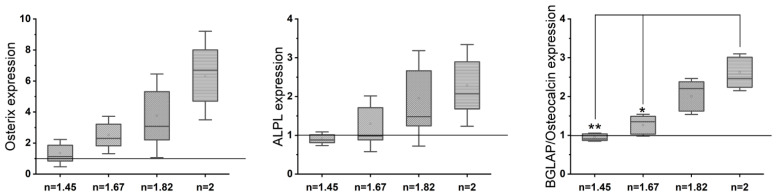
Gene expression analysis (rt-PCR) shows an upregulation of osteogenic markers on SiON_x_ as compared to the baseline control. The increase in the N/O ratio increases the osteogenic marker expression. * *p* < 0.05 and ** *p* < 0.01 present the significance compared to the normal TCP as a control.

**Figure 6 antioxidants-13-00189-f006:**
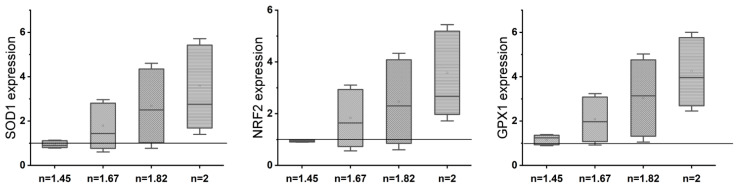
Gene expression analysis (rt-PCR) shows an upregulation of antioxidant markers in SiON_x_ as compared to the baseline control. The increase in the N/O ratio increases the antioxidant marker expression.

**Figure 7 antioxidants-13-00189-f007:**
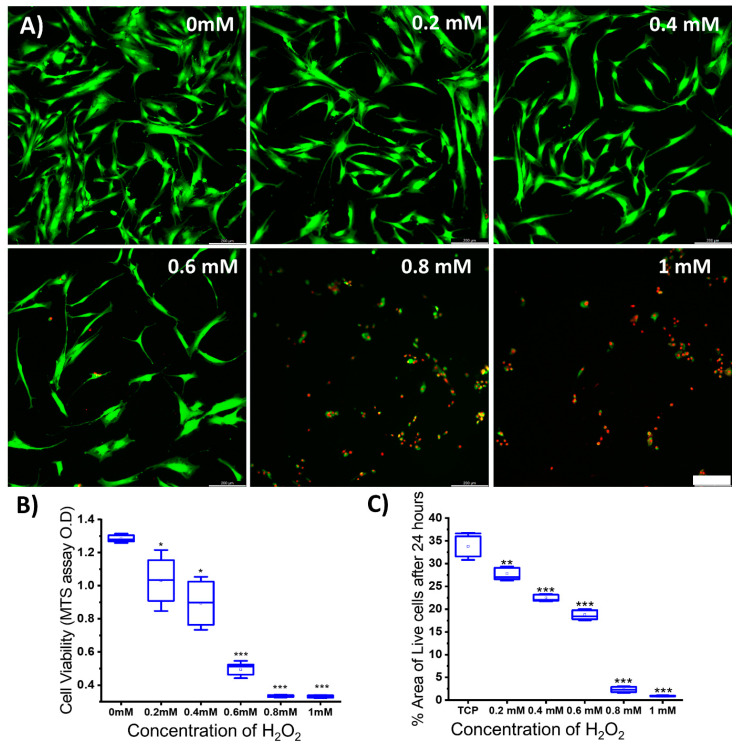
(**A**) Cell viability tested under various hydrogen peroxide concentrations; (**B**) MTS assay; (**C**) quantification of live cells using ImageJ software. * *p* < 0.05, ** *p* < 0.01, and *** *p* < 0.001 present the significance compared to the normal control of 0 mM H_2_O_2_ or the TCP as a control. Scale bar = 200 µm.

**Figure 8 antioxidants-13-00189-f008:**
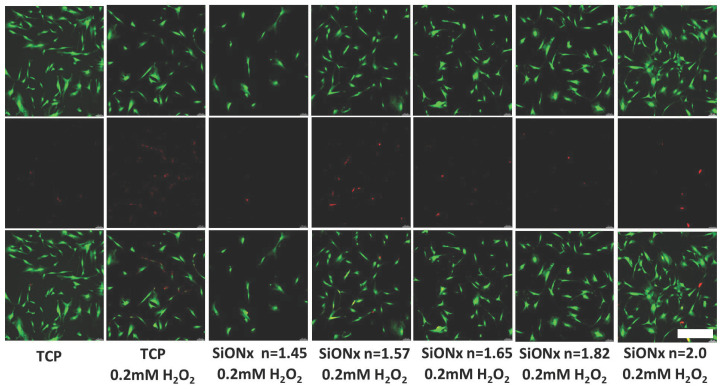
Cell proliferation on various SiON_x_ chemistries under toxic oxidative stress. Green color (stained by Calcein AM) represents the live cells, and red color (stained by Ethidium homodimer) represents the dead cells. Scale bar = 200 µm.

**Figure 9 antioxidants-13-00189-f009:**
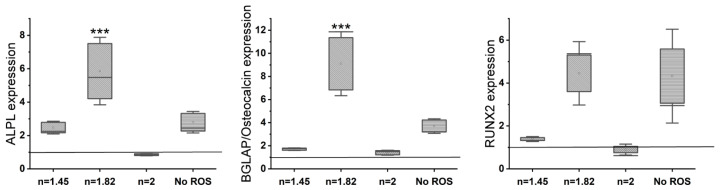
Gene expression analysis (rt-PCR) shows a significant upregulation of osteogenic markers in SiON_x_ *n* = 1.82 as compared to the baseline control in an oxidative stress environment. The osteogenic marker expression is not significantly different from the baseline with no oxidative stress involvement. *** *p* < 0.001 presents the significance compared to the control.

**Figure 10 antioxidants-13-00189-f010:**
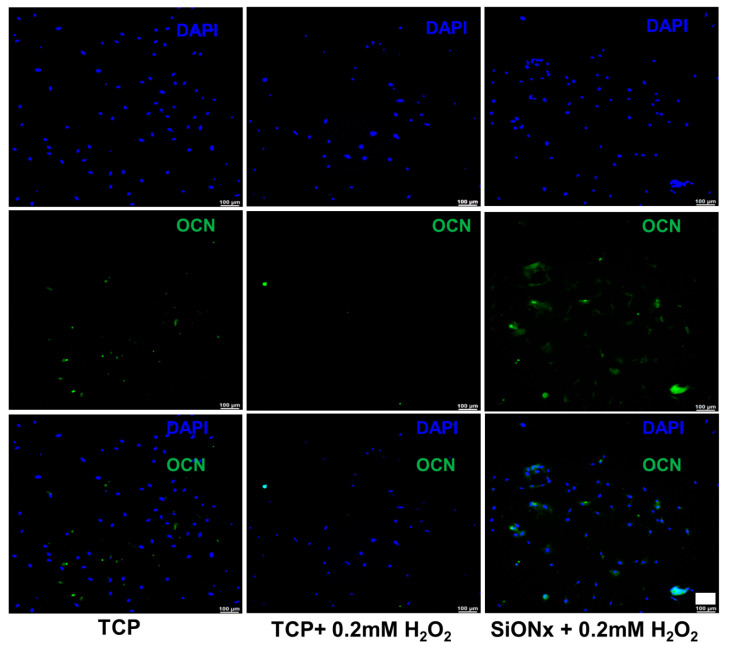
IHC staining representing DAPI (blue) and osteocalcin (green) on MSCs after 7 days of osteogenic differentiation. Overexpression of the osteogenic marker on the selected SiON_x_ (*n* = 1.82) surface vs. the control is evident. Scale bar = 100 µm.

**Figure 11 antioxidants-13-00189-f011:**
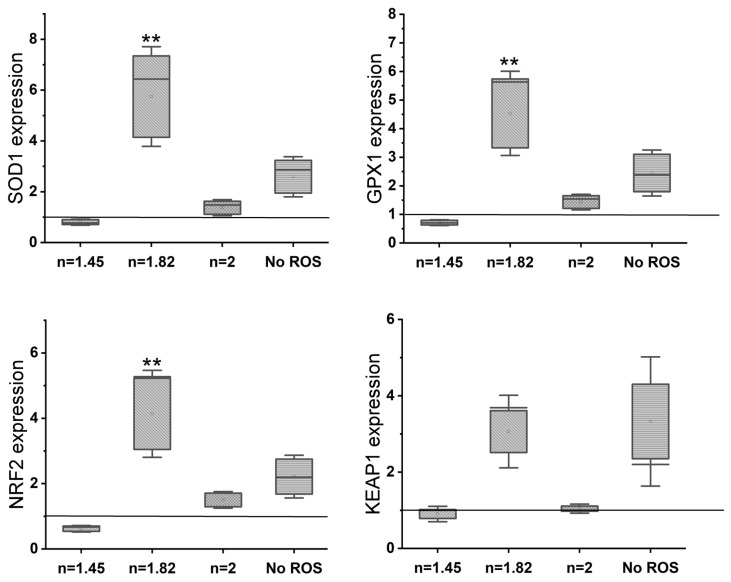
Gene expression analysis (rt-PCR) shows a significant upregulation of antioxidant markers in SiON_x_ *n* = 1.82 as compared to the baseline control in an oxidative stress environment. ** *p* < 0.01 presents the significance compared to the baseline control (TCP).

**Figure 12 antioxidants-13-00189-f012:**
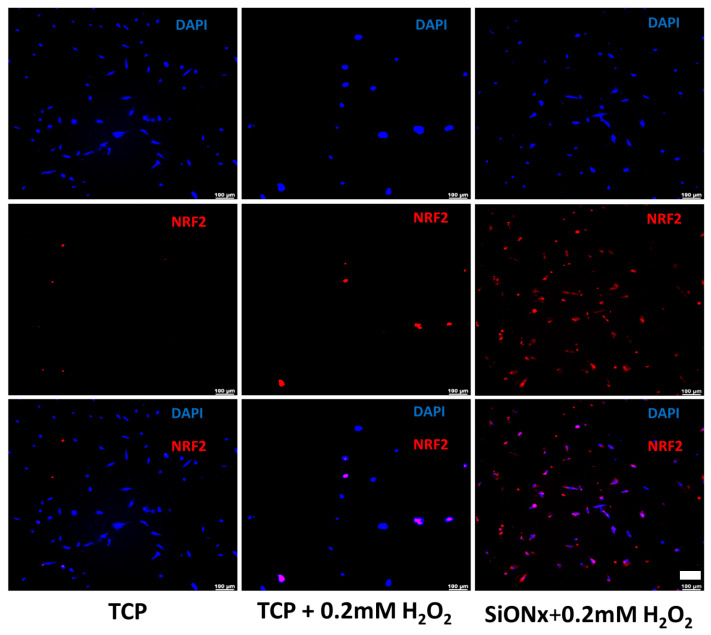
IHC staining representing DAPI (blue) and NRF2 (red) on MSCs after 7 days of osteogenic differentiation. There is a significant increase in the antioxidant NRF2 activity on the selected SiON_x_ (*n* = 1.82) surface when compared to the control. Scale bar = 100 µm.

**Figure 13 antioxidants-13-00189-f013:**
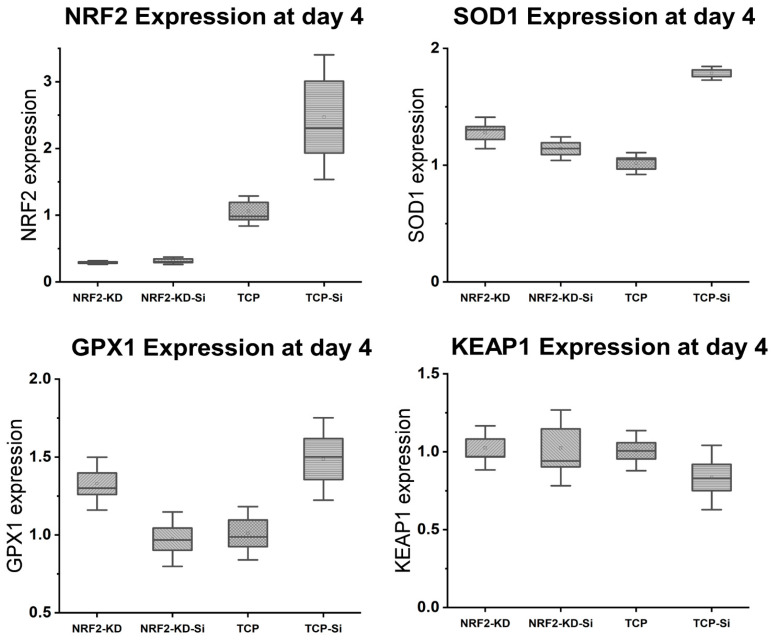
NRF2 KD blocks the Si ion protective effect on the antioxidant expression.

**Figure 14 antioxidants-13-00189-f014:**
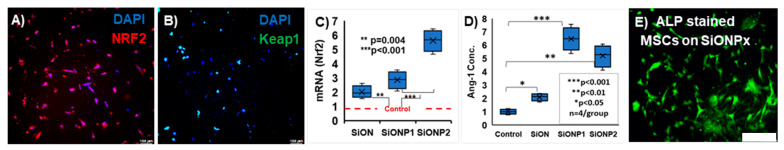
SiON_x_ and SiONP_x_ enhanced NRF2 and angiopoietin (ANG1) vs. control. (**A**) DAPI (blue)/NRF2 (red): IHC image shows cytosolic NRF2 expression in MSCs (exposed to ROS) on SiON_x_ coatings. NRF2 was stained with polyclonal anti-NRF2 Ab, followed by Alexa-594 secondary Ab, while the nucleus was counterstained with DAPI. (**B**) DAPI (blue)/Keap1 (green): IHC image shows nuclear localization of Keap1 in MSCs (exposed to ROS) on SiON_x_ coatings. SiONP_x_ (SiONP1: P/N = 0.9, SiONP2: P/N = 1.0) enhanced NRF2 activity (**C**) and ANG1 (**D**) vs. SiON_x_. (**E**) Alkaline phosphatase (ALP)-stained differentiated MSCs on SiONP_x_-Ti. * *p* < 0.05, ** *p* < 0.01, *** *p* < 0.001 present the significance compared to the control. Scale bar = 200 µm.

**Table 1 antioxidants-13-00189-t001:** Gas flow rates, deposition rate, and refractive index for SiON_x_ or SiONP_x_ layers deposited by PECVD.

Sample	Coating	Gas Flow Rate (sccm)				Deposition Rate (nm/min)	Refractive Index (*n*)
		15% SiH_4_/Ar	N_2_O	N_2_	NH_3_		
**1**	**SiON_x_**	24	160	225	50	62.5	1.45
**2**		24	155	225	50	59.5	1.57
**3**		24	16	225	50	44.5	1.65
**4**		24	3	225	50	41.0	1.82
**5**		24	0	225	50	36.5	2.0
		**15%SiH_4_/2%PH_3_/Ar**	**N_2_O**	**N_2_**	**NH_3_**	**Deposition Rate**	**Refractive index (*n*)**
**6**	**SiONP_x_**	24	16	225	50	38.5	1.68

## Data Availability

The data presented in this study are available on request from the corresponding author.

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
