# Peer review of "SiONx Coating Regulates Mesenchymal Stem Cell Antioxidant Capacity via Nuclear Erythroid Factor 2 Activity under Toxic Oxidative Stress Conditions"

_antioxidants, 2024, doi:10.3390/antiox13020189_

Round 1

Reviewer 1 Report (New Reviewer)

Comments and Suggestions for Authors

The article #antioxidants-2778237 entitled “SiONx Coating Regulates Mesenchymal Stem Cell Antioxidant Capacity via NRF2 Activity Under Toxic Oxidative Stress Condition” has been submitted for publication in the journal Antioxidants (MDPI).

The research deals with the use of SiONx thin films deposited on silicon substrate by plasma enhanced chemical vapor deposition (PECVD) technique. These surface layers are used to improve bioactivity, bacteriostatic activity, and reduce oxidative stress to the surface of bone implants. More particularly, the effect of several different concentrations in oxygen and nitrogen is explored. Several in vitro studies are conducted involving human mesenchymal stem cells (MSC) and bacterial colonies in contact with the deposited SiONx coatings. The authors describe proliferation, osteogenic differentiation, and antioxidant activity for 1-7 days.

The results show improved biological behaviors due to the action of the SiONx thin films that could be used to improve bone regeneration in vivo in an oxidative stress environment.

In my opinion, this article contains interesting results that deserve to be published. However, some points needs to be clarified. I recommend minor corrections of the manuscript according to the following comment:

 - in the whole manuscript, the letter “x” must be subscripted in “SiONx”.

 - the studied thin films are deposited on silicon wafers which is not a regular material for bone implant applications. Are the properties of these SiONx thin films the same if deposited on metal or polymer substrates? This point needs to be discussed in the manuscript.

 - The study describes the impact of different concentrations in oxygen and nitrogen, but the different thin films are characterized in the article by their refractive index, an optical property with no interest for the bone implant applications. Some chemical characterizations of the films must be added to highlight the impact of the oxygen and nitrogen concentrations on the biological properties of the films.

Author Response

Response to reviewer comments

We thank the reviewers and the academic editor for the consideration and the very comprehensive review of our manuscript. We have modified the manuscript according to reviewer’s comments plus added consideration from the authors.

Responses to Reviewer 1

  1. In the whole manuscript, the letter “x” must be subscripted in “SiONx”.

Response: We thank the reviewer for catching this correction, we have revised accordingly and made the changes.

  1. The studied thin films are deposited on silicon wafers which is not a regular material for bone implant applications. Are the properties of these SiONx thin films the same if deposited on metal or polymer substrates? This point needs to be discussed in the manuscript.

Response: The PECVD films do not change their properties whether deposited on Si wafers or on Ti implants/ materials. We have numerous publications showing this effect as given in response to comment 3 below. In summary, the films are always amorphous and have the same optical/ chemical bulk and surface properties no matter the substrate. We have added short paragraph in the introduction to Cleary describe this, please see page 2 lines 81-86.

  1. The study describes the impact of different concentrations in oxygen and nitrogen, but the different thin films are characterized in the article by their refractive index, an optical property with no interest for the bone implant applications. Some chemical characterizations of the films must be added to highlight the impact of the oxygen and nitrogen concentrations on the biological properties of the films.

Response: We have extensively characterized these biomaterials for their chemical structure and surface morphology as well as O/N ratio effect on the chemical structure. We have edited the manuscript to mention this, please see page 5, lines 248-255. For these characterization before and after coating, please refer to the following citations below. They also include characterization of hydroxyapatite nucleation and surface chemical and mechanical characterization of these coatings, composition analysis by XPS and NRA-RBS, and other properties that we have previously determined that biological bioactivity:

  • K Awad, S Young, P Aswath, and V Varanasi, Interfacial adhesion and surface bioactivity of anodized titanium modified with SiON and SiONP surface coatings. Surfaces and Interfaces, 2022. 28: p. 101645. 10.1016/j.surfin.2021.101645
  • A Ilyas, NV Lavrik, HKW Kim, PB Aswath, and VG Varanasi, Enhanced Interfacial Adhesion and Osteogenesis for Rapid “Bone-like” Biomineralization by PECVD-Based Silicon Oxynitride Overlays. ACS Applied Materials & Interfaces, 2015. 7(28): p. 15368-15379. 10.1021/acsami.5b03319
  • Varanasi, Venu G., Ilyas, Azhar, Velten, Megen F., Shah, Ami, Lanford, William A., Aswath, Pranesh B. Role of Hydrogen and Nitrogen on the Surface Chemical Structure of Bioactive Amorphous Silicon Oxynitride Films. The Journal of Physical Chemistry B, 2017, 121, 38, 8991–9005, https://doi.org/10.1021/acs.jpcb.7b05885

Reviewer 2 Report (New Reviewer)

Comments and Suggestions for Authors

In the manuscript titled SiONx Coating Regulates Mesenchymal Stem Cell Antioxidant Capacity via Activity Under Toxic Oxidative Stress Condition, Ahuja et al. investigated the influence of titanium surface coating using SiONx films with various N/O ratios on the antimicrobial activity against MRSA and cytocompatibility towards MSCs. The rationale of the study is well-justified, the obtained results are described properly, however, several major issues appeared and must be explained, otherwise, the presented manuscript should not be published.

1. The physicochemical characterization of materials is missing. Please provide SEM-EDX micrographs of the implant surface before and after coating. To prove the amorphous nature of the coating, please provide XRD results. FTIR or XPS results would be also valuable to prove the chemical composition of the surfaces.

2. Do the Authors think that increased expression of antioxidant markers in the presence of proposed SiONx films (Figure 3) is a desirable feature? Did the Authors consider the possible phenomenon that increased expression results from a defensive mechanism against pro-oxidative species that are released from the film? Please discuss it.

3. How was the phosphate added to the structure of SiONx? How were the P contents and P/N molar ratios confirmed? Please add the proper description in the Materials and methods section as well as expand your aim of the study. In my opinion, the presented study is large and this data together with Figures 13 and 14 should be provided in supplementary materials.

4. The Authors stated that “The increasing nitrogen content in the SiONx chemistry leads to the formation of highly oxidative protonated radicals, which will then result in bacterial inactivation”. It is very confusing because it means that proposed coatings exert proinflammatory properties and induce oxidative stress as free radicals.

5. Did the Authors consider the potential degradation/neutralization of H2O2 in the presence of SiONx? The in vitro studies in which the activity of H2O2 in the presence and absence of SiONx should be carried out.

6. Ion release studies must be carried out to prove the statement mentioned in lines 315-323. Presently the results are not sufficient. The hydroxyapatite formation in simulated body fluid must be provided to prove the apatite formation.

7. In the case of antimicrobial activity assay, both sessile and planktonic bacteria should be investigated. Was the concentration of bacteria inoculum 105 CFU/mL or 10^5 CFU/mL? SEM images of the implants’ surface after the test would be valuable. How were the materials sterilized before in vitro studies?

8. In Figure 3, the results should be presented as a % of control instead of OD values. Are the results presented in Figure 3 related to the MTT assay? If yes, it should be absorbance instead of OD.

9. The cell viability under physiological and oxidative stress conditions should be compared quantitatively using MTT assay. Does the residual H2O2 not interfere with the reagents of the MTT test?

10. The abstract section is too long. The structure of the presented abstract is more suitable for a conference abstract instead of a scientific paper. Moreover, what does it mean fixative device – should that be a fixation device?

11. What does it mean sccm? Please explain how the was refractive index measured and how it correlated with the N/O ratio. Please provide the molar ratio of N/O in each coating used.

12. In the introduction, the role of NRF2 in the bone regeneration process should be briefly described.

13. The name and mechanism of action of the live/dead bacteria assay kit must be provided.

14. I do not see the supplementary data attachment.

15. Figure 2 resolution is very low.

16. The results presented in Figure 7 should be transferred to supplementary materials.

17. In Figures 9 and 11, why are the results for all investigated types of SiONx not presented?

18. In Figures 10 and 12, the n value of SiONx used must be added.

19. What type of Si, N, O ions do you mean (line 335)? Please prove it.

20. The third paragraph in the discussion section is not a discussion and should be transferred to the introduction section. Moreover, I do not understand the importance of the fourth paragraph in the discussion section.

Comments on the Quality of English Language

The English language is understandable.

Author Response

Response to reviewer comments

We thank the reviewers and the academic editor for the consideration and the very comprehensive review of our manuscript. We have modified the manuscript according to reviewer’s comments plus added consideration from the authors.

Responses to Reviewer 2

  • The physicochemical characterization of materials is missing. Please provide SEM-EDX micrographs of the implant surface before and after coating. To prove the amorphous nature of the coating, please provide XRD results. FTIR or XPS results would be also valuable to prove the chemical composition of the surfaces.

Response: We have extensively characterized these biomaterials for their chemical structure and surface morphology as well as O/N ratio effect on the chemical structure. We have edited the manuscript to mention this, please see page 5, lines 248-255. For these characterization before and after coating, please refer to the following citations below. They also include characterization of hydroxyapatite nucleation and surface chemical and mechanical characterization of these coatings, composition analysis by XPS and NRA-RBS, and other properties that we have previously determined that biological bioactivity:

  • K Awad, S Young, P Aswath, and V Varanasi, Interfacial adhesion and surface bioactivity of anodized titanium modified with SiON and SiONP surface coatings. Surfaces and Interfaces, 2022. 28: p. 101645. 10.1016/j.surfin.2021.101645
  • A Ilyas, NV Lavrik, HKW Kim, PB Aswath, and VG Varanasi, Enhanced Interfacial Adhesion and Osteogenesis for Rapid “Bone-like” Biomineralization by PECVD-Based Silicon Oxynitride Overlays. ACS Applied Materials & Interfaces, 2015. 7(28): p. 15368-15379. 10.1021/acsami.5b03319
  • Varanasi, Venu G., Ilyas, Azhar, Velten, Megen F., Shah, Ami, Lanford, William A., Aswath, Pranesh B. Role of Hydrogen and Nitrogen on the Surface Chemical Structure of Bioactive Amorphous Silicon Oxynitride Films. The Journal of Physical Chemistry B, 2017, 121, 38, 8991–9005, https://doi.org/10.1021/acs.jpcb.7b05885

  • Do the Authors think that increased expression of antioxidant markers in the presence of proposed SiONx films (Figure 3) is a desirable feature? Did the Authors consider the possible phenomenon that increased expression results from a defensive mechanism against pro-oxidative species that are released from the film? Please discuss it.

Response: We thank the reviewer for this view. Yes, we agree with the reviewer. Activating the NRF2 as an antioxidant pathway in oxidative stress conditions, which is predominant in trauma and bone fracture cases, is desirable. Nrf2 plays a crucial defensive role in regulating the antioxidant response, please see page 2 lines 54-64.  Regarding the increased expression, we have already found in our prior publication that ionic Si releases from these films. Silicon ions are cations and play a reductive role electrochemically. Thus, Our working hypotheses are that (1) increased charge transfer from surface Si4+ valence state catalyzes ROS reduction (Table 2), (2) surface Si4+ may enhance NRF2 activity due to Keap1 sensitivity to 2+ and 4+ cations [47] thereby allowing NRF2 release. This was mentioned in page 11 line 360-365 and table 2 for the proposed chemical equations. Please refer to the below publication for more information:

  • A Ilyas, T Odatsu, A Shah, F Monte, H Kim, P Kramer, P Aswath, and V Varanasi, Amorphous Silica: A New Antioxidant Role for Rapid Critical-Sized Bone Defect Healing. Advanced healthcare materials, 2016. 50(17): p. 2199-2213. PMC6635139
  • A Ilyas, M Velton, A Shah, F Monte, HKW Kim, PB Aswath, and VG Varanasi, Rapid Regeneration of Vascularized Bone by Nanofabricated Amorphous Silicon Oxynitrophosphide (SiONP) Overlays. J Biomed Nanotechnol, 2019. 15(6): p. 1241-1255. PMID 31072432

  • How was the phosphate added to the structure of SiONx? How were the P contents and P/N molar ratios confirmed? Please add the proper description in the Materials and methods section as well as expand your aim of the study.

Response:  We have edited the materials and methods section and table 1 to describe how we fabricate the SiONPx coatings. The P content was quantified by HR-SEM-EDX and A ZAF quantification method was used to determine the P/N, N/Si, O/Si and P/Si atomic ratios in the coatings. Regarding the method of phosphate addition and these data, please refer to the following publications.

  • A Ilyas, M Velton, A Shah, F Monte, HKW Kim, PB Aswath, and VG Varanasi, Rapid Regeneration of Vascularized Bone by Nanofabricated Amorphous Silicon Oxynitrophosphide (SiONP) Overlays. J Biomed Nanotechnol, 2019. 15(6): p. 1241-1255. PMID 31072432
  • K Awad, S Young, P Aswath, and V Varanasi, Interfacial adhesion and surface bioactivity of anodized titanium modified with SiON and SiONP surface coatings. Surfaces and Interfaces, 2022. 28: p. 101645.

  • The Authors stated that “The increasing nitrogen content in the SiONx chemistry leads to the formation of highly oxidative protonated radicals, which will then result in bacterial inactivation”. It is very confusing because it means that proposed coatings exert proinflammatory properties and induce oxidative stress as free radicals.

Response:  We thank the reviewer for catching this statement. We previously characterized the coating surfaces (citation below) and as the nitrogen containing species increases with no pro-oxidant species present. We have corrected this text and apologize for the mistake. Please see corrected section at page 12, line 377-380.

  • A Ilyas, NV Lavrik, HKW Kim, PB Aswath, and VG Varanasi, Enhanced Interfacial Adhesion and Osteogenesis for Rapid “Bone-like” Biomineralization by PECVD-Based Silicon Oxynitride Overlays. ACS Applied Materials & Interfaces, 2015. 7(28): p. 15368-15379. PMC6508966

  • Did the Authors consider the potential degradation/neutralization of H2O2 in the presence of SiONx? The in vitro studies in which the activity of H2O2 in the presence and absence of SiONx should be carried out.

Response: Yes, this data is shown in figure 8. TCP group is a tissue culture plate that has just normal medium, while group TCP+0.2mM H2O2 is the normal media plus the H2O2 without SiONx. Then 5 different groups of SiONx coatings were tested. Please refer to Figures 7, 8 and 9. 

  • Ion release studies must be carried out to prove the statement mentioned in lines 315-323. Presently the results are not sufficient. The hydroxyapatite formation in simulated body fluid must be provided to prove the apatite formation.

Response: We have already published results on in vitro Si ion release and surface HA formation on the coatings. We have edited the text to clearly mention that and refer to the references. These citations are below:

  • K Awad, S Young, P Aswath, and V Varanasi, Interfacial adhesion and surface bioactivity of anodized titanium modified with SiON and SiONP surface coatings. Surfaces and Interfaces, 2022. 28: p. 101645.
  • A Ilyas, NV Lavrik, HKW Kim, PB Aswath, and VG Varanasi, Enhanced Interfacial Adhesion and Osteogenesis for Rapid “Bone-like” Biomineralization by PECVD-Based Silicon Oxynitride Overlays. ACS Applied Materials & Interfaces, 2015. 7(28): p. 15368-15379. PMC6508966
  • A Ilyas, M Velton, A Shah, F Monte, HKW Kim, PB Aswath, and VG Varanasi, Rapid Regeneration of Vascularized Bone by Nanofabricated Amorphous Silicon Oxynitrophosphide (SiONP) Overlays. J Biomed Nanotechnol, 2019. 15(6): p. 1241-1255. PMID 31072432

  • In the case of antimicrobial activity assay, both sessile and planktonic bacteria should be investigated. Was the concentration of bacteria inoculum 105 CFU/mL or 10^5 CFU/mL? SEM images of the implants’ surface after the test would be valuable. How were the materials sterilized before in vitro studies?

Response: MRSA forms a biofilm on the surface of a foreign device which is a community of sessile bacteria. The Staphylococci account for up to two thirds of all pathogens isolated from orthopedic implant infections. This bacterium was used to evaluate the antimicrobial effect of our coatings according to ASTM and ISO standards. We have edited the text to describe this, please see page 5, line 224-233.  We corrected the concentration to 10^5 CFU/mL. We added the sterilization methods. All samples were cleaned using 100% ethanol followed by dry heat sterilization prior each experiment [46]. Unfortunately, we don’t have available SEM pictures of the bacteria, but live/dead fluorescent images are provided for all groups and at three time points.

  • In Figure 3, the results should be presented as a % of control instead of OD values. Are the results presented in Figure 3 related to the MTT assay? If yes, it should be absorbance instead of OD.

Response: We thank the reviewer for these recommendations, we have made the requested changes in Figure 3.

  • The cell viability under physiological and oxidative stress conditions should be compared quantitatively using MTT assay. Does the residual H2O2 not interfere with the reagents of the MTT test?

Response: These data is presented in Figure 7 (Normal) and Figure 8 (Under oxidative stress) using MTS assay not MTT. This new MTS assay is known as CellTiter 96® AQueous One Solution Cell Proliferation Assay. It is a colorimetric method for determining the number of viable cells in proliferation or cytotoxicity assays. The CellTiter 96® AQueous One Solution Reagent contains a novel tetrazolium compound [3-(4,5-dimethylthiazol-2-yl)-5-(3-carboxymethoxyphenyl)-2-(4-sulfophenyl)-2H-tetrazolium, inner salt; MTS] and an electron coupling reagent (phenazine ethosulfate; PES). PES has enhanced chemical stability, which allows it to be combined with MTS to form a stable solution and minimize the interference with medium or chemical substances. Also, we have run media containing H2O2 without cells to account for any possible background, this is subtracted from all results. We added this clarification in page 4, line 165.

  • The abstract section is too long. The structure of the presented abstract is more suitable for a conference abstract instead of a scientific paper. Moreover, what does it mean fixative device – should that be a fixation device?

Response: We have reduced the abstract size. In the craniomaxillofacial field, fixative device and fixation device are used interchangeably. We have changed it to “fixation device” for improved readability.

  • What does it mean sccm? Please explain how the was refractive index measured and how it correlated with the N/O ratio. Please provide the molar ratio of N/O in each coating used.

Response: The use of “sccm” means standard centimeters cubed per minute, we have added this to the text. Refractive index and its relationship to composition is given in our prior papers as given below.

  • A Ilyas, T Odatsu, A Shah, F Monte, H Kim, P Kramer, P Aswath, and V Varanasi, Amorphous Silica: A New Antioxidant Role for Rapid Critical-Sized Bone Defect Healing. Advanced healthcare materials, 2016. 50(17): p. 2199-2213. PMC6635139
  • A Ilyas, M Velton, A Shah, F Monte, HKW Kim, PB Aswath, and VG Varanasi, Rapid Regeneration of Vascularized Bone by Nanofabricated Amorphous Silicon Oxynitrophosphide (SiONP) Overlays. J Biomed Nanotechnol, 2019. 15(6): p. 1241-1255. PMID 31072432

  • In the introduction, the role of NRF2 in the bone regeneration process should be briefly described.

Response:  We have added new paragraph to the introduction, please see page 2, line 54-64.

  • The name and mechanism of action of the live/dead bacteria assay kit must be provided.

Response: We have added these details. Page 5, line 238.

  • I do not see the supplementary data attachment.

Response: We have attached the supplementary data file to this submission.

  • Figure 2 resolution is very low.

Response: We made sure that all figures are at least 300 dpi or more.

  • The results presented in Figure 7 should be transferred to supplementary materials.

Response: We appreciate the reviewer suggestions but given the importance of this figure to the rest of the results in the remainder of the manuscript and the request given by the reviewer above, it seems that this figure should remain in the main body of the manuscript as it appears to add important results for the manuscript.

  • In Figures 9 and 11, why are the results for all investigated types of SiONx not presented?

Response: We observe that SiONx with the refractive index of n= 1.82 enhances bioactivity and maximizes the antioxidant effect against the oxidative stresses, exhibiting [Si-Si]-[Si-O]-[Si-N] bonds as compared to SiONx n=2 (which is nitrogen-rich, SiNx) and the low nitrogen content SiONx n=1.45, SiOx like structure. Thus, these three groups were used for theses two experiments.  

  • In Figures 10 and 12, the n value of SiONx used must be added.

Response: We have made these changes. Thank you.

  • What type of Si, N, O ions do you mean (line 335)? Please prove it.

Response: This was a typo. Only Si ions are released as we previously reported. We apologize for the mistake and have deleted O and N from the sentence.

  • The third paragraph in the discussion section is not a discussion and should be transferred to the introduction section. Moreover, I do not understand the importance of the fourth paragraph in the discussion section.

Response: We agree with the reviewer and have moved paragraph 3 to the introduction section. Paragraph 4 is being used to explain why the antioxidant effect occurs from the biomaterials.

Round 2

Reviewer 2 Report (New Reviewer)

Comments and Suggestions for Authors

Dear Authors,

I am satisfied by the improvements you made.
The paper can now be published.

This manuscript is a resubmission of an earlier submission. The following is a list of the peer review reports and author responses from that submission.

Round 1

Reviewer 1 Report

Comments and Suggestions for Authors

This article explores the impact of the oxygen to nitrogen ratio in SiONx coatings on the osteogenic capacity of bone cells. The authors investigate the role of the NRF2 antioxidant pathway in regulating this effect and demonstrate that the coatings with a higher oxygen to nitrogen ratio promote greater osteogenic capacity. The study provides new insights into the potential of SiONx coatings as a biomaterial for bone tissue engineering applications.

The title needs to be improved to match the manuscript’s content well.

The abstract is long and should be cut it down in half.

This article presents research on the impact of the oxygen to nitrogen ratio in SiONx coatings on the osteogenic capacity of bone cells and the role of the NRF2 antioxidant pathway in regulating this effect. While the study provides new insights into the potential of SiONx coatings as a biomaterial for bone tissue engineering applications, the novelty of this work is not clear.

It does not make sense to study the ALP activity at days 1, 4 and 7. The biomineralization should be investigated at a longer culture time.

Here are some tips that might help the authors

ALP activity assays are commonly used to measure the osteogenic differentiation of bone cells. The timing of the assay can vary depending on the specific study design and research question, but common cell culture time points for ALP activity assay include:

Day 7-14: Early stages of osteogenic differentiation

Day 14-21: Mid-stages of osteogenic differentiation

Day 21-28: Late stages of osteogenic differentiation

The reviewer noted that the list of references has to be updated as some of them since 1989 and 2001.

I recommend consulting recent primary literature, reviewing articles, and conducting a literature search using databases such as PubMed to find the most up-to-date and relevant references for your research. It is important to note that the timing of ALP activity assay can vary depending on the specific study design and research question, therefore, it is crucial to consult the most recent literature in order to have a clear and updated understanding of the field.

Here you have suggested references that would help the authors to update their references and support their work

https://www.sciencedirect.com/science/article/abs/pii/S2772950822005283

https://www.sciencedirect.com/science/article/pii/S0272884222039293

https://www.sciencedirect.com/science/article/pii/S1549963417301739

https://www.sciencedirect.com/science/article/abs/pii/S0169433219334208

https://pubs.acs.org/doi/abs/10.1021/acsami.2c00918

https://www.sciencedirect.com/science/article/pii/S1385894718306120

https://www.mdpi.com/2079-4991/11/11/2799

https://www.sciencedirect.com/science/article/pii/S0264127521006493

Author Response

Please see attached files of the response and the revised manuscript. 

Reviewer 2 Report

Comments and Suggestions for Authors

Thank you for the interesting content. Indeed, we are faced with a certain percentage of the consequences and complications of fractures, the treatment of which were with usage of metallic osteosynthesis. The possibility of various coatings using that have not only bactericidal and bacteriostatic activity but enhance the biological activity of the fixing device in terms of reparative osteogenesis is an interesting and promising R&D activity. The manuscript left the most pleasant impressions, has a scientific novelty and the prospect of continuing research. At the same time, I want to note some shortcomings that have a technical nature, can be easily corrected and do not affect the quality of the material as a whole.

So:

1. Inaccuracy in the caption and interpretation of the figure 1. Probably an inaccuracy, since the text indicates that SiNOx n=2.0 significantly reduces the number of colonies after 12 hours, while from the images that I recommend labeling a) , b) and c), it follows that SiNOx n=2.0 reduces the number of colonies after 24 and 48 hours.

2. In figure 1, I recommend removing the captions Р=…….., which overload the diagrams.

3. On the diagrams, where there are symbols *, **, ***, it is necessary to give a decoding between which groups significant differences were recorded.

4. The materials and methods indicate that the analysis of alkaline phosphatase activity was carried out colorimetrically with the Abcam kit (lines 162-163), and the results (lines 259, 260) indicate the determination of activity by ELISA. Clarification is necessary, since it is known that the activity of this enzyme is determined by a unified colorimetric method with p-nitrophenyl phosphate, and the concentration of the enzyme is determined by ELISA. In our opinion, it is necessary to exclude the reference to the ELISA method from the text or provide explanations.

5. It is necessary to transfer the diagrams from the "Discussion" section to the "Results" section, with the appropriate numbering.

In conclusion, I would like to wish the authors success in their research work!

Author Response

Please see the attached files of the response and the revised manuscript. 
